# Adaptive Coping Strategies at the Time of COVID-19: The Role of Social and General Trust

**DOI:** 10.3390/ijerph20156512

**Published:** 2023-08-03

**Authors:** Anna Rita Graziani, Lucia Botindari, Michela Menegatti, Silvia Moscatelli

**Affiliations:** 1Department of Communication and Economics, University of Modena and Reggio Emilia, Viale Allegri 9, 42121 Reggio Emilia, Italy; 2SAIS Europe, Johns Hopkins University, Via Andreatta 3, 40126 Bologna, Italy; lbotindari@jhu.edu; 3Department of Psychology, University of Bologna, Viale Berti Pichat 5, 40126 Bologna, Italy; m.menegatti@unibo.it (M.M.); silvia.moscatelli@unibo.it (S.M.)

**Keywords:** general trust, social trust, coping strategies, COVID-19, pandemic, wellbeing

## Abstract

Research in the field shows the crucial role of trust in the functioning of many aspects of social life, especially when dealing with emergencies. We report the results of a study (*N* = 883) carried out in Italy during the first phase of the COVID-19 crisis to assess whether and how social trust (i.e., trust in those who have the authority and responsibility for making decisions, such as the Italian government, the regional government, the Civil Protection, the European Union, the Catholic Church, and the scientific community) and general trust (i.e., trust in the trustworthiness of other individuals, such as Italians and humankind) are associated with the employment of coping strategies in dealing with the challenges of the pandemic. Overall, the results highlight that trust in different authorities and institutions and trust in other human beings are positively associated with the adoption of adaptive coping strategies such as seeking social support, positive reinterpretation and growth, problem-solving orientation, and transcendent orientation. In contrast, they are negatively related to the adoption of maladaptive coping strategies such as avoidance. These findings point out the benefits of various types of trust in helping individuals deal with crises, such as the COVID-19 pandemic.

## 1. Introduction

The outbreak of the COVID-19 pandemic dramatically influenced the personal lives of many people worldwide and led them to face unprecedented challenges. This new acute respiratory syndrome created havoc for societies, economies, communities, and individuals [1]. National governments imposed urgent measures limiting individual freedom and required behaviors that went against shared social norms, such as maintaining physical distance from the members of in-groups, including family and friends [2]. The success of the fight against the virus greatly depended on citizens’ adherence to such restrictive policies and their intentions to engage in protective behaviors, such as getting vaccinated.

The pandemic gave rise to many questions about the virus, its origin, and its ways of transmission, some of which were left unanswered for several months. Moreover, it was accompanied by a flourishing of conspiracy theories offering explanations for the spreading of the pandemic and its management by governments [3,4]. When laypeople lack knowledge of hazards such as COVID-19, they need to rely on “trusted others” to assess the risk and to implement correct behaviors. Trust represents a “social glue” in societies and is a crucial element for social functioning, as it helps individuals act in a complex environment and facilitates the pursuit of collective goals [5,6]. Trust in those who have the authority and responsibility for making decisions, such as scientists and government agencies (the so-called “social trust” [7]), as well as general trust—that is, the belief that “most people are trustworthy most of the time” [8]—can play a critical role in facing uncertain and complex situations such as the COVID-19 pandemic. Indeed, studies conducted during previous pandemics such as H1N1 or Ebola and during the COVID-19 pandemic highlighted that trust in governments and other institutions, citizens, and science can favor the acceptance of prescribed behaviors and engagement in prosocial behaviors [9,10,11].

The present study, which was conducted in Italy during the first months of the COVID-19 outbreak and the national lockdown, examines the association between social and general trust and individuals’ adoption of coping strategies to face the coronavirus pandemic. Specifically, it focuses on trust in the different institutions and governments that were in charge of managing the crisis or that played an important role in supporting people (such as the Catholic Church) as well as generalized trust in other Italians and humankind.

### 1.1. Social and General Trust during the Outbreak of COVID-19

Trust is a critical element for social functioning at both micro and macro levels: it increases interpersonal cooperation and makes social and political institutions more effective and economic activities more efficient [12,13]. Trusting other people, groups, or institutions can also be perceived as a risk [14], but individuals accept this vulnerability based upon their positive expectations of the intentions or behavior of who they are trusting [15].

As mentioned, research on risk management suggested distinguishing between social and general trust [5]. Social trust can be defined as the willingness to rely on those who have the responsibility for taking actions related to technology, environment, medicine, public health, and safety [7]. This kind of trust is particularly relevant in situations where the individual has insufficient knowledge or an insufficient technical background to make decisions and take action: trusting scientists and government agencies represents a way to reduce the complexities and fears that people face.

Research showed that social trust influences, for example, the perceived risks concerning pesticides, nuclear power, and artificial sweeteners [7] and the likelihood of accepting a referendum to site a hazardous waste disposal facility [16] or irradiated food [17]. In the health domain, studies on H1N1 influenza found that trust in the government, medical organizations, and the media plays a crucial role in promoting the acceptance of the recommended behavior to control the spread of the disease [11,18]. Moreover, social trust positively influences the intention to be vaccinated against this virus inflection [19].

The literature on social capital [20] suggested that general trust is related to “how much someone trusts people whom one meets for the first time” [8] (p. 788). This belief in the benevolence of human nature is not limited to a particular group or organization, but it is a default expectation of others’ trustworthiness [21]. General trust is essential for social cooperation and economic exchanges [12]. Moreover, it plays an important role in digitalized societies, where people can easily form new social relationships that are no longer bound by interpersonal social networks [22].

Research focusing on the COVID-19 pandemic examined trust from two perspectives. On one side, studies pointed out an increase in social trust as a consequence of the pandemic. Evidence from European countries demonstrated that the rise of the pandemic increased social trust, particularly trust in the government because of “the rally-around-the-flag-effect” [23,24]. According to Schraff [25], anxiety related to and fear of the contagion and its consequences rallied people around political institutions to increase security in an unmoored situation. In addition, several studies showed that the outbreak of COVID-19 increased trust in science or scientists [26]. Little attention was, instead, devoted to the impact of COVID-19 on general trust. A study conducted in Norway found no overall increase in generalized trust in the early phase of the pandemic, which was, however, higher for individuals who had a direct experience with the disease compared to those who perceived themselves as at risk or were worried about the pandemic [27]. Research [28] found a generalized increase in interpersonal trust during the COVID-19 pandemic compared to the pre-pandemic level among the Italian population and increased levels of trust in strangers among people who caught COVID-19. Different results for social and political trust emerged in China, where greater exposure to COVID-19 risks significantly decreased general and political trust among the adult population [29].

On the other side, a consistent corpus of the literature is focused on the impact of trust on the measures to reduce infection. Researchers [8] found that individuals who trusted the government and the pharmaceutical industry regarding SARS-CoV-2 tended to perceive more risks and showed more acceptance of containment measures than participants with lower social trust. Moreover, individuals with higher levels of general trust perceived less health risk and were less inclined to accept the related measures, probably because they believed in the benevolence of others and, therefore, found it difficult to perceive others as risk factors of contagion.

The positive influence of social trust found further support. A higher level of government trust and trust in its ability to control COVID-19 were significantly associated with greater compliance with protective behaviors such as frequent handwashing, avoidance of crowded spaces, and social isolation or quarantine [30,31]. Similarly, greater trust in science and scientists increased individuals’ tendency to follow the COVID-19 prevention guidelines [32] and to accept the lockdown [33]. A recent study [10] found that trust in governments, science, and fellow citizens was a better predictor of compliance with prescribed behaviors than information related to the actual threat of the virus. In this case, differently from Siegrist and colleagues [8], general trust positively influenced the acceptance of the containment measures. However, studies [34,35] highlighted the apparently paradoxical effects of generalized trust, which was related to higher compliance with vaccination but lower support for nonpharmaceutical interventions such as wearing masks. This might be due to people’s beliefs about the ability of others to respect social distancing or to the fact that those with a high level of general trust fail to acknowledge that all people can pose a health risk.

Other studies [4] highlighted the role of trust in mediating the effects of identification with Italians and the beliefs of conspiracy theories on the wish for a strong leader to deal with the pandemic. Individuals’ identification with the national group was related to a lower wish for a strong leader through the mediation of higher trust. In contrast, the belief in conspiracy theories was associated with decreased trust and, through it, with an increased need for a strong leader. However, in Moscatelli and colleagues’ study [4], it was impossible to disentangle the effect of social and general trust since the authors employed a combined measure.

Considering the role of trust in fostering well-being, Paolini and colleagues [36] reported that Italians’ trust in social (e.g., journalists and the Civil Protection) and political actors (e.g., the National Health System and the prime minister) was positively related to well-being during the first phase of the COVID-19 pandemic, and this effect was mediated by the identification with Italians and humankind. Moreover, Roccato and colleagues [37] found that increased trust in political, super partes, and international institutions positively influenced well-being by reducing anxiety, collective angst, and anger via the mediation of participants’ perceived control over their lives. Finally, Moscatelli and colleagues [9] reported that Italians’ higher identification with their country and the EU was related to increased levels of trust in the Italian and European governments, which, in turn, accounted for greater expectations that the pandemic crisis would eventually have positive outcomes, by improving Italian politics, the EU, and humankind. However, as far as we know, no research has so far examined the impact of social and general trust on predicting adaptive coping strategies in response to pandemic challenges, adaptive coping strategies that are crucial to maintaining individuals’ physical and psychological well-being during such a crisis.

### 1.2. The Role of Coping Strategies Dealing with COVID-19

Coping strategies, that is, the set of cognitive, emotional, and behavioral strategies that individuals use to master, tolerate, reduce, or minimize especially stressful events [38], are crucial to buffer adversities due to the virus. Coping strategies can be divided into approach and avoidance coping strategies [39]. The approach strategies are represented by actions aimed at directly alleviating the problem (e.g., seeking emotional support and planning to resolve and reduce stressors). In contrast, avoidance strategies aim at distancing oneself from the problem (e.g., withdrawing from others, denying reality, and suppressing thoughts and emotions regarding the stressor). Traditionally, avoidance coping was considered an unsuccessful strategy, especially in the long term [39,40,41,42]. Research showed that different strategies fulfill different functions [38]: emotion-focused strategies (e.g., denial, venting, and emotional support) serve to manage and reduce emotional distress, whereas problem-focused strategies (e.g., informational support and active coping) aim to solve or remove the source of stress.

Of interest here, Foà and colleagues [43], based on the work of Carver and colleagues [44], proposed and validated a measure of coping strategies based on five critical dimensions. Among them, three dimensions are related to active coping strategies: (a) problem-solving orientation, that is, the tendency to search for information and plan solutions while suppressing other activities that can interfere with problem solving; (b) positive attitude and reinterpretation, intended as the tendency to accept and reinterpret events in order to transform them in opportunities of growth; and (c) avoidance, which represents the attempt to ignore and deny the stressful event. The other two strategies seem less focused on the stressor and are related to seeking the help of others and of higher entities, that is, (d) seeking social support (the tendency to turn to others for emotional support and requesting advice) and (e) transcendent orientation, which is the tendency to seek comfort in religion [43].

In general terms, research pointed out that the way individuals react to stressful events can have long-term psychological and health effects. For instance, using avoidance strategies is likely related to depression and anxiety [45,46,47]. Conversely, engaging in cognitive reappraisal and problem-solving strategies is conducive to better mental health outcomes and well-being [48,49,50]. Similarly, an approach coping strategy was found to be related to a higher psychological health and a better quality of life [51,52].

The different roles of approach and avoidance coping strategies emerged during the COVID-19 pandemic. Approach strategies such as positive reinterpretation and planning strategies were protective against anxiety and depressive symptoms, whereas avoidance coping strategies predicted higher levels of anxiety and depression [53]. Similarly, a study conducted in Austria during the 4 weeks of the COVID-19 lockdown [54] revealed that coping strategies predicted a set of mental health indicators. Specifically, individuals who engaged in positive thinking, active stress coping, and social support strategies reported a higher psychological life quality, higher levels of well-being, and lower levels of stress, depression, anxiety, and insomnia. In particular, positive thinking was the strongest predictor, followed by social support. This result aligns with previous evidence as well as with other studies conducted during the pandemic, which found that positive thinking was related to lower depressive symptoms and lower distress in Italian health care professionals [55] and in the Greek population [56]. However, a study conducted among Lebanon adults reported that both approach and avoidance coping strategies predicted higher general well-being during the COVID-19 pandemic [57]. The positive role played by avoidance strategies might be explained by the specific nature of the stressor. Indeed, it is possible that people benefitted from taking a break from the threat and the several issues related to the pandemic, taking psychological distance from them while regaining energy to more directly address them [57].

Interestingly, for our purpose, Fluharty and colleagues [58] explored predictors of coping strategies during the COVID-19 pandemic in UK adults. First, they found clear differences due to demographic and social background factors. Problem-focused and emotion-focused coping strategies were used more by women, older people, and more educated and higher income people, whereas they were less influenced by factors such as living conditions. Women, people with more education and a higher income, younger people, and people living with others were also more likely to search for social support, whereas women and people from more disadvantaged groups (i.e., people with lower educational attainment and a lower socioeconomic position, people with mental health conditions, people in overcrowded living condition, and people who were lonelier) showed a tendency to engage in avoidance strategies. Since religion was not measured independently from emotion-focused coping, no specific conclusion can be drawn in this respect. This study allowed for the understanding of the variations in responses to the COVID-19 pandemic, while highlighting which groups could benefit from interventions to improve their coping abilities, at the same time the focus on demographic variables and living conditions left more psychosocial variables in the shadows. Thus, it is important to examine how trust in different groups and institutions, as well as generalized trust in others, relates to the reliance on different coping strategies during a long-term stressor such as the COVID-19 pandemic.

### 1.3. Research Overview

The present study investigated whether social and general trust were associated with different coping strategies to face the COVID-19 pandemic. It was carried out during the first months of the pandemic, when Italy was the worst-hit country after China and the first Western country to enter a national lockdown, and there were no proven treatments for COVID-19. In this situation of great uncertainty, people had to turn to political and scientific authorities for information and guidance to prevent the spread of the infection.

Regarding social trust, we focused our attention on trust in several institutions that, at different levels, were involved in managing the pandemic and containment policies: the Italian government, the government of one’s region, the Civil Protection, and the European Union. Furthermore, we considered trust in the scientific community and the Catholic Church. In fact, the scientific community was highly salient at the time, since scientists were moving at a record speed to find effective treatments and create new vaccines and were often required to comment upon the pandemic on mass and social media. For its part, the Catholic Church played a crucial role in supporting people facing emergencies with existential reasons and psychological resources [59].

The efficacy of the containment measures depended on people’s adherence to such restrictions. General trust concerned the belief that Italian citizens and humankind would do their best to face the COVID-19 pandemic, for instance, by respecting the rules imposed by the government and scientific authorities.

In general terms, we expected that social and general trust were related to the adoption of adaptive coping strategies when facing uncertainty during pandemic times. Specifically, we considered the following adapting strategies: (a) seeking social support, which implies seeking information and support from significant others; (b) positive reinterpretation and growth, which implies the acceptance, containment, and positive interpretation of the situation; (c) problem-solving orientation, which implies focusing on the problem, suppressing competitive activities, planning strategies to overcome the problem, and acting; (d) transcendence orientation, which implies turning to God and praying for help and support. Moreover, we expected that trust would be negatively associated with the recourse to the avoidance strategy, which implies negation and behavioral and mental detachment from the problem.

## 2. Materials and Methods

### 2.1. Method

#### 2.1.1. Participants

A total of 1146 Italian participants were recruited through social networking sites and snowball sampling and volunteered for the study. Inclusion criteria were the following: living in Italy, being 18 years or older, and filling out the informed consent form. While age was registered at the end of the questionnaire, participants were informed that the questionnaire was intended for adults only (i.e., ≥18 years old) in the initial consent statement. In addition, participants were excluded if they did not complete the questionnaire. Four participants were excluded as they did not provide consent, while further 259 did not complete the questionnaire, leaving a final sample of 883 participants (639 women, 244 men; Mage = 38.33 years; SD = 14.86 years; range 18–79 years; 541 living in Northern Italy and 342 living in Central and Southern Italy). A total of 229 participants were categorized as having had “personal experience with COVID-19”, as they reported having contracted the coronavirus (60) and/or indicated that a family member or a close friend had contracted it (195).

#### 2.1.2. Procedure

The Bioethical Committee of the University of Bologna approved the project. The questionnaire was administered via Qualtrics between 15 and 30 April 2020 and included further measures that were published elsewhere [9,60] and are not described in this paper. Respondents were first presented with measures of trust in different institutions and people; then, they filled in the measure of coping. Afterwards, respondents were asked whether they, a member of their family, or a close friend had contracted the coronavirus and indicated their religion (1 = I do not believe in any religion; 2 = Catholic religion; 3 = Jewish religion; 4 = Islamic religion; 5 = Buddhism; 6 = Other religion; 7 = I believe in a Higher Entity, but I do not identify with any religion). The great majority of respondents (535; 60.6%) reported to be Catholic; 202 respondents (22.9%) indicated that they did not believe in any religion; 13 respondents (1.5%) indicated Buddhism, 37 respondents (4.2%) choose “Other religion”, and 96 (10.8%) indicated that they believed in a Higher Entity without identifying with any religion. Since we were interested in trust in the Catholic Church as predictor of coping, we categorized the respondents distinguishing between non-Catholic (= 0; *n* = 348) and Catholic (= 1; *n* = 535).

Afterward, respondents reported their political orientation on an 11-point left–right scale (0 = extremely left, and 10 = extremely right; M = 4.00, SD = 2.20) and provided demographic information, including their region of residence. Since, at the time of data collection, Italy’s northern regions were more severely hit by the coronavirus in comparison with central and southern regions, we categorized respondents’ places of residence in northern regions and central–southern regions.

#### 2.1.3. Measures

Trust. Participants read “Thinking to the following groups and institutions, please indicate how much they are able to face the COVID-19 pandemic.” The instructions were followed by a list of groups and institutions. For social trust, the institutions were “the government of my region”; “the Italian government”; “the Civil Protection”; “the European Union”; “the Catholic Church”; and “the scientific community”. For general trust, “the Italian citizens” and “humankind” were considered. The answers were given on a 5-point scale ranging from 1 (not at all) to 5 (very much).

Coping strategies. We adopted the 25-item version of the Coping Orientation to the Problems Experienced—New Italian Version (COPE-NIV; [43]), which is derived from Carver’s and colleagues [44] Coping measure (see also [61]). The COPE-NIV measure refers to five macro strategies of coping: seeking social support (5 items), positive reinterpretation and growth (6 items), problem-solving orientation (5 items), avoidance (5 items), and transcendent orientation (4 items). The answers were given on a 7-point scale ranging from 1 (not at all) to 7 (very much).

We conducted a Confirmatory Factor Analysis (CFA) with Mplus 8.4 to determine whether the items fitted the structure of the measure, as reported by [43]. To examine the model fit, we used various indices [62]: the comparative fit index (CFI) and the Tucker–Lewis index (TLI), which should exceed 0.90 to be considered acceptable, and the root-mean-square error of approximation (RMSEA) and the standardized root-mean-square residual (SRMR), which should be less than 0.08 [63]. The fit of the model was initially not completely adequate according to the described fit criteria: CFI = 0.850; TLI = 0.831; RMSEA = 0.076, 95% CI [0.072, 0.079]; SRMR = 0.079. Based on the inspection of the modification indexes, we deleted 6 items (1 item from seeking social support; 3 items from positive reinterpretation and growth; 2 items from avoidance). After deleting these items, fit indices resulted as acceptable: CFI = 0.932; TLI = 0.919; RMSEA = 0.061, 95% CI [0.056, 0.066]; SRMR = 0.052. The reliability indexes were acceptable: seeking social support (α = 0.68), positive reinterpretation and growth (α = 0.70), problem-solving orientation (α = 0.71), avoidance (α = 0.60), and transcendent orientation (α = 0.96).

#### 2.1.4. Data Analysis

All the analyses were run with IBM SPSS Statistics 23, except for the CFA on the measure of coping strategies, which (as mentioned earlier) was conducted with Mplus 8.4. To examine the hypothesized associations between social and general trust and the coping strategies, we conducted a series of bootstrapped hierarchical regression analyses with 5000 resamples, separately considering each coping strategy as the outcome variable. In Model 1, we accounted for the variability due to socio-demographic variables, personal experience with COVID-19, and political orientation. Specifically, in Model 1, we included age, gender (0 = man; 1 = woman), place of residence (0 = Northern Italy; 1= Central–Southern Italy), personal experience with COVID-19 (0 = no; 1 = yes), political orientation, and Catholic religion (0 = no, 1 = yes). To analyze the relationship between different forms of trust and coping strategies, in Model 2, we added trust measures. Bivariate Pearson correlations were also conducted to determine linear relationships among all continuous variables.

To gain an overview of participants’ levels of social and general trust in different institutions and groups, we ran pairwise *t*-tests comparing different trust measures. To adjust for multiple comparisons and decrease the likelihood of committing a Type I error (falsely rejecting the null hypothesis), we applied the Bonferroni correction for multiple testing and adopted more stringent levels of significance. This quite conservative correction consists of dividing the nominal significance level of the α test (e.g., α = 0.05) by the number of tests [64]. For pairwise comparisons, the highest number of tests is 27 (for the trust measures); accordingly, the significance level was set at *p* = 0.002 (0.05/27). For independent-sample comparisons, since we compared the groups with respect to 13 variables, the significance level was set at *p* = 0.004 (0.05/13).

## 3. Results

### 3.1. Descriptive Statistics

Table 1 reports the means and standard deviations of the trust and the coping strategies measures for the total sample and as a function of gender, place of residence (northern regions; central-southern regions) personal experience with COVID-19 infection (no; yes), and Catholic religious affiliation (no; yes).

### 3.2. Social and General Trust

First, a series of pairwise *t*-tests was run to compare respondents’ level of social and general trust in different institutions and groups. Respondents turned out to trust the scientific community more than all the other groups and institutions considered, *t*s(882) > 11.85, *p*s < 0.001, *d*s > 0.41. Respondents trusted the Civil Protection more than all the other institutions and groups (except for the scientific community), *t*s(882) > 7.40, *p*s < 0.001, *d*s > 0.25. Trust in the regional government was higher than trust in the Italian government, the EU, the Catholic Church, Italian citizens, and humankind, *t*s(882) > 7.31, *p*s < 0.001, *d*s > 0.25. Respondents reported higher trust in the Italian government compared to the EU and humankind, *t*s(882) > 5.63, *p*s < 0.001, *d*s > 0.19, while they reported trusting humankind more than Italian citizens, the Catholic Church, and the EU, *t*s(882) > 4.71, *p*s < 0.001, *d*s > 0.16. Trust in Italian citizens was higher than trust in the EU or the Catholic Church, *ts*(882) > 7.94, *p*s < 0.001, *d*s > 0.27, whereas trust in the EU and trust in the Catholic Church scored lower than trust in all the other groups and institutions and did not differ from each other, *t*(882) = −0.64, *p* = 0.521.

Second, we conducted a series of independent sample *t*-tests to examine whether trust in the considered institutions and groups differed as a function of respondents’ gender, place of residence, personal experience with COVID-19, and religion (Catholic vs. non-Catholic). The findings revealed that female respondents showed higher trust than men did in the Civil Protection, *t*(881) = 4.79, *p* < 0.001, *d* = 0.36, and the Catholic Church, *t*(881) = 3.30, *p* < 0.001, *d* = 0.25. Since, as mentioned, a significance level of *p* < 0.004 was set to adjust for multiple comparisons, no other comparisons between men and women can be considered significant, *p*s > 0.008. Respondents living in the north of Italy reported higher trust in the regional government, *t*(881) = 8.19, *p* < 0.001, compared to those who lived in the central–southern regions. No other comparisons between respondents living in Northern vs. Central–Southern Italy were significant, *p*s > 0.011. There were no significant differences due to personal experience with COVID-19, *p*s > 0.060. Finally, respondents who identified as Catholic reported higher trust in the Catholic Church, *t*(881) = 10.49, *p* < 0.001, *d* = 0.72, compared to non-Catholic respondents. The analyses showed no other effects due to religion, *p*s > 0.013.

### 3.3. Coping Strategies

Pairwise *t*-tests highlighted some differences in the recourse to the various coping strategies. Positive reinterpretation and growth turned out to be the most relevant coping strategy, with scores being higher than those of all other strategies, *t*s(881) > 31.44, *p*s < 0.001, *d*s > 1.08. Seeking social support scored higher than avoidance and transcendent orientation, *t*s(882) > 18.01, *p*s < 0.001, *d*s > 0.62, while it scored lower than problem-solving orientation, *t*(882) = −5.42, *p* < 0.001, *d* = 0.19. Finally, the scores of problem-solving orientation were higher than those of avoidance and transcendent orientation, *t*s(882) > 21.50, *p*s < 0.001, *d*s > 0.74, while the scores of avoidance were lower than those of transcendent orientation, *t*(882) = −5.97, *p* < 0.001, *d* = 0.21.

Independent sample *t*-tests revealed that women scored lower than men on the avoidance strategy, *t*(881) = −3.47, *p* < 0.001, *d* = −0.26, while they scored higher than men on all other strategies, *t*s (881) > 3.47, *p*s < 0.001, *d*s > 0.26. Participants living in Central–Southern Italy reported higher levels of the transcendent orientation strategy compared to participants from Northern Italy, *t*(882) = 7.21, *p* < 0.001, *d* = 0.50. No other comparison between the two areas of residence reached the statistical level of significance set for this analysis, *t*s(882) < 2.45, *ps* > 0.014, *d*s < 0.17. Regarding experience with COVID-19, those who had no personal experience made a greater recourse to the avoidance strategy, *t*(882) = 3.95, *p* < 0.001, *d* = 0.30, with a nearly significant effect on the transcendent orientation strategy, *t*(882) = 2.87, *p* = 0.004, *d* = 0.22. There was no other significant effect for experience with COVID-19, *t*s (882) < −1.97, *p*s > 0.049, *d*s < 0.15. Finally, participants who identified as Catholic scored higher than non-Catholic participants on the transcendent orientation strategy, *t*(882) = 19.49, *d =* 1.34. No other comparison between Catholic and non-Catholic participants reached statistical significance, *t*s (882) < 2.39, *p*s > 0.008, *d*s < 0.16.

### 3.4. Hierarchical Regression Analyses

Table 2 presents bivariate correlations among the measures of trust and coping strategies.

Overall, the results confirmed the existence of significant correlations between coping strategies and the dimensions of social and general trust. However, it is noteworthy to point out a different pattern of correlation between the different forms of social trust and the coping strategies. Seeking social support was positively correlated with trust in the Italian government, in the Civil Protection, in the EU, in the Catholic Church, and in the scientific community. No significant correlation emerged between seeking social support strategy and trust in the regional government. Positive reinterpretation and growth was correlated with all the different social trust measures, while problem-solving orientation positively correlated only with trust in the Civil Protection, in the Catholic Church, and in the scientific community. Trust in the regional government, in the Italian government, in the Civil Protection, in the Catholic Church, and in the scientific community was negatively correlated with the avoidance strategy. The analysis did not reveal any significant correlation between avoidance and trust in the EU. Transcendent orientation was positively correlated with trust in the Catholic Church and negatively correlated with trust in the regional government and in the EU. Trust in Italian citizens and in humankind, as the dimension of general trust, was positively correlated with seeking social support, positive reinterpretation, problem-solving orientation, and transcendent orientation. Regarding the avoidance coping strategy, the analyses revealed only a negative correlation with trust in Italian citizens.

As explained before, to test for the associations among different forms of trust and coping strategies, we conducted a series of hierarchical regression analyses, whereby age, gender, place of residence, personal experience with COVID-19, political orientation, and Catholic religion were entered in Model 1, and the trust measures were entered in Model 2. Coping strategies were entered as outcome variables. Table 3 shows the results of these models.

The analyses revealed, for Model 1, the significant effects of age and gender on the seeking social support strategy. Being younger and being female were associated with a stronger reliance on this strategy. When the trust measures were entered in Model 2, stronger trust in the Catholic Church and in the scientific community were significantly associated with a greater tendency to seek support to face the pandemic.

Second, in Model 1, the positive reinterpretation and growth strategy was positively associated with being female and living in Central–Southern Italy. In Model 2, reliance on this strategy was negatively related to being Catholic and trusting the EU. Conversely, trust in the Catholic Church, in the scientific community, and in humankind was positively associated with reliance on the positive reinterpretation and growth strategy, while trust in the EU was negatively associated with such a strategy.

In Model 1, the analysis concerning the problem-solving orientation strategy revealed, again, positive associations with being female and living in Central–Southern Italy. Respondents who reported a personal experience with COVID-19 also scored higher on such a strategy. In Model 2, trust in the Catholic Church and trust in humankind were significantly related to problem-solving orientation.

In Model 1, gender, age, experience with COVID-19, and political orientation had significant associations with the avoidance strategy. Specifically, being female, being older, and having had personal experience with COVID-19 were associated with a lower reliance on the avoidance strategy. Right-wing political orientation was, instead, positively associated with this strategy. In Model 2, trust in the EU was positively associated with the reliance on the avoidance strategy, which was, instead, negatively related to trust in the Catholic Church and trust in the scientific community.

Finally, in Model 1, being older, being female, living in Central–Southern Italy, being right-wing-oriented, and being Catholic were associated with a greater recourse to the transcendent orientation strategy. In Model 2, trust in the regional government, in the Italian government, and in the EU was negatively related to this strategy, which was instead positively associated with trust in the Catholic Church and humankind.

## 4. Discussion

The present research aimed to verify the role of trust in helping individuals to face the strains and uncertainties due to COVID-19. In particular, we focused on social trust, that is, the confidence that authorities and institutional organizations would be able to manage the pandemic, and general trust, which represents the confidence that other human beings would do their best to limit the spread of disease. To reach this goal, we conducted a study during the first phase of the pandemic evolution in Italy, focusing on different aspects of social and general trust regarding the adoption of individual coping strategies against COVID-19. Specifically, we considered four adaptive strategies of coping, that is, seeking social support, positive reinterpretation and growth, problem-solving orientation, and transcendent orientation, and one maladaptive form of coping, that is, avoidance. Overall, the findings supported the general expectation that social trust and general trust were associated with individuals’ ability to employ different coping strategies.

First, the seeking social support coping strategy underlines the importance of significant others and one’s own social network for information seeking and understanding and for emotional support [65]. This adaptive strategy implies the perception of being valued by others and of being part of a social network [66] and concerns humans’ fundamental need for belongingness [67]. Our results confirmed the importance of seeking social support and showed that trust in the Catholic Church and in the scientific community was associated with a stronger tendency to seek information and emotional support from others. It is noteworthy that despite having different scopes and often being considered to be in conflict with each other [68], both institutions promoted an adaptive coping strategy that helps individuals focus on the problem’s solution and the relief from emotional stress.

It should also be noted that, in general terms, the tendency to seek social support seems to decrease with age. A possible explanation may rely on the peculiar situation due to the pandemic, since the lockdown challenged the way of maintaining social connections. Digital and online technologies became crucial to maintain interpersonal relationships, but, at the same time, they prevented older adults from staying connected with others since some older people were unable or reluctant to use technology, leaving them vulnerable to social isolation [69].

Positive reinterpretation and growth is a coping strategy that refers to the acceptance, containment, and positive reframing of stressful events [61]. It is considered an adaptive strategy that helps individuals face negative emotions by construing a stressful situation in positive terms and focusing on one’s own emotional growth. This strategy was strongly employed by our participants. Concerning the effect that social trust exerted on this coping strategy, the results confirmed the positive association between trust in the Catholic Church and in the scientific community. The higher the trust in these different institutions, the more participants considered the pandemic as an opportunity for personal growth.

Interestingly, trust in the European Union was negatively associated with this coping strategy. We suspect that, at the very beginning of the pandemic, Italians’ trust in the EU might have been under strain due to the lack of a joint and coordinated reaction by European countries, as well as by the awareness that Italy was the only EU country to be severely hit by COVID-19 [70,71]. This result can be interpreted in terms of compensatory processes: the decrease in trust in Europe could have enhanced participants’ need to restore control over uncertain and unpredictable events by positively reframing stressful events.

Moreover, identifying oneself as Catholic had a negative relation with the positive reinterpretation of COVID-19 events. This result seems to contradict the role of trust in the Catholic Church. A possible explanation relies on the unique Italian situation, especially in comparison to other European nations. A vast majority of Italians continue to consider themselves Catholic, even if this affiliation is now expressed in a variety of forms, with different levels of intensity, and, in many cases, it is characterized by contradictions and feelings of ambivalence [72]. Moreover, compared to the past, regularly practicing Catholics are now a minority in Italy, even if they are far more numerous than in other European nations. In our opinion, the measure of trust in the Catholic Church captured participants’ tendency to identify with the institution and consider it as an epistemic authority, while the self-definition of being Catholic is a broad category characterized by pluralistic points of view. Such a distinction can help understand the seemingly contradictory findings concerning trust in the Catholic Church and religiosity.

Finally, our findings highlighted that general trust—specifically, trust in humankind—was associated with the positive reinterpretation coping strategy. Indeed, the confidence that other human beings would be able to properly act to limit the spread of the pandemic increased the positive interpretation of this dramatic event.

Problem-solving orientation is a problem-focused coping strategy characterized by the employment of active and interpretative strategies to change or eliminate the underlying causes of stress via individual behavior [61]. It is considered an adaptive coping strategy that helps individuals focus on the root of the problem. Our results showed that this strategy was particularly employed by people living in the center–south of Italy and by those who had experience with COVID-19. Again, trust in the Catholic Church and trust in humankind were positively related with the employment of this coping strategy: the higher the trust in the Catholic Church institution and in other human beings was, the more our participants used planning and active strategies to face the pandemic’s strains.

Avoidance is a coping strategy aimed at escaping stressful situations, experiences, or difficult thoughts and feelings rather than dealing with them [61]. It is generally considered as a maladaptive form of coping because it does not address the sources of stress, and it tends to increase stress and anxiety when overused. The regression analysis revealed that age and having experience with COVID-19 were negatively associated with the use of the avoidance strategy: older participants and those who had personal experience with the disease tended to limit the employment of this strategy. Moreover, the analysis showed interesting results for social trust: considering the Catholic Church and the scientific community as a resource to fight the pandemic represented a protective factor that limited the tendency to escape from the problem. Analogous to the positive interpretation strategy, trust in the European Union did not seem to help individuals employ adaptive coping strategies: the higher the trust in the EU was, the more participants relied on avoidance-oriented coping strategies to face the pandemic’s stressful events.

Finally, transcendent orientation is an emotion-focused coping strategy that refers to turning to religion to find comfort through praying or meditating. The literature showed that in Italy there was an increase in religiousness during the COVID-19 crisis: people derived more comfort in religious activities during the hard times of the pandemic [73,74,75]. Our findings showed that place of residence, political orientation, and religious beliefs were positively related to the employment of this strategy. Participants who lived in the center–south, who had right-wing beliefs, and who defined themselves as Catholic tended to refer to religion to find relief from COVID-19’s strains and uncertainties. Interestingly, the results concerning social trust highlighted a different pattern of results. Trust in civil authorities such as the regional government, the Italian government, and the European Union reduced the use of this coping strategy, while trust in the Catholic Church was positively associated with its employment. Concerning general trust, our findings showed that trust in humankind seems to lead to relying on religion and prayer to face a stressful situation.

An important strength of the current study is that we considered different aspects of social trust. In particular, different authorities that played an important role in managing the pandemic were considered: several national and European political actors, the Catholic Church, and the scientific community. The results from the correlation analyses showed that the different dimensions of social trust correlated differently with the coping strategies. For example, the higher the trust in the Civil Protection, in the Catholic Church, and in the scientific community was, the higher the individuals’ employment of problem-solving orientation coping strategies. The regional, Italian, and European authorities did not seem to play an important role in helping individuals to plan strategies to overcome the situation. This was probably due to the peculiar situation, since, in the first phase of the pandemic, local, national, and European authorities might have been under strain due to the lack of information and certainty about the infection, and citizens considered the Civil Protection, the Catholic Church, and the scientific community as more trustworthy epistemic authorities. At the same time, all the social trust dimensions, except for trust in the EU, negatively correlated with the avoidance coping strategy. The higher the trust in these authorities was, the lower the employment of this maladaptive coping strategy was. As previously pointed out, at that time, when Italy was the first European nation dealing with COVID-19, the lack of clear support could diminish trust in the European Union. These results and the evidence gathered from the regression analyses showed that social trust dimensions are not all alike in helping individuals to adopt adaptive coping strategies. Moreover, they highlight that social trust cannot be taken for granted, but the level of trust increases or decreases as a consequence of the authorities’ ability to deal with dramatic events.

Interestingly, our results showed significant gender differences in coping mechanisms, with women using more adaptive coping strategies than men. According to the literature, women tend to use more emotion-focused coping, whereas men use more problem-focused coping strategies, when dealing with stressful events (e.g., [76]). Our results showed that, irrespective of the focus of the coping strategies (emotional vs. problem-solving), our female participants were able to employ adaptive coping strategies more often than male participants. Furthermore, women used avoidant coping strategies less often than men.

### Limitations and Future Directions

Some limitations of the present study should be acknowledged. Firstly, the participants’ recruitment adopted the snowball sampling strategy, which is not based on a random selection of the sample. Therefore, the study’s sample did not reflect the actual pattern of the general Italian population. In addition, the online administration of the questionnaire could also have undermined the sample representativeness. Secondly, the cross-sectional design limited the possibility of solidly establishing cause-and-effect relationships between the examined variables. Future studies could investigate in depth the cause-and-effect relationships between the different forms of trust and the employment of adaptive coping strategies.

## 5. Conclusions

Overall, these findings contribute to the literature on the antecedents of coping strategies as well as to the growing evidence concerning individuals’ responses to an unprecedented global event such as the COVID-19 pandemic. In particular, they highlight how, in the first phase of the pandemic—when uncertainty about the causes and the better ways to deal with the situations was at its most—feelings of trust with different institutions as well as of general trust in others played a protective role and positively influenced individuals’ coping. Given that adopting adaptive coping strategies—such as problem solving, positive reinterpretation, transcendent orientation, and seeking social support—was found to be conducive to a higher level of well-being and fewer depression and anxiety symptoms [48,49,50], interventions to improve resilience and mental health during stressful situations should consider the important role played by social, political, and religious institutions. In general terms, it is important that practitioners and other professionals are aware that social trust and general trust in others are critical factors in sustaining people during tough situations.

Our results showed that individuals’ feelings that they can rely on institutions as well as on the scientific community as resources for dealing with crises can be crucial in dramatic situations. Of course, political, social, religious, and scientific agencies should build the basis for social trust before an emergency event occurs, because unreliable authorities may not be trusted by citizens. Consequently, citizens could be distrustful of the adoption of authorities’ recommended behaviors [8]. National and European political leaders as well as the scientific community should be aware of the importance of nurturing citizens’ social trust by employing effective policy and clear communication strategies.

## Figures and Tables

**Table 1 ijerph-20-06512-t001:** Descriptive statistics as a function of gender and place of residence.

	Total	Gender		Place of Residence		Experience with COVID-19		Catholic Religion	
		Women	Men	North	Center/South	No	Yes	No	Yes
Variables	M (SD)	M (SD)	M (SD)	M (SD)	M (SD)	M (SD)	M (SD)	M (SD)	M (SD)
*Social Trust*									
1. Regional Government	3.19 (0.87)	3.24 (0.83)	3.07 (0.95)	3.37 (0.81)	2.90 (0.87)	3.15 (0.85)	3.29 (0.91)	3.23 (0.91)	3.17 (0.84)
2. Italian Government	2.96 (0.82)	2.96 (0.82)	2.93 (0.82)	2.97 (0.82)	2.95 (0.83)	2.93 (0.81)	3.05 (0.84)	2.99 (0.83)	2.94 (0.82)
3. Civil Protection	3.43 (0.84)	3.50 (0.81)	3.21 (0.85)	3.48 (0.80)	3.34 (0.88)	3.39 (0.84)	3.52 (0.82)	3.41 (0.83)	3.43 (0.84)
4. European Union	2.34 (0.84)	2.37 (0.83)	2.28 (0.85)	2.38 (0.85)	2.25 (0.82)	2.34 (0.84)	2.35 (0.82)	2.43 (0.83)	2.29 (0.83)
5. Catholic Church	2.37 (1.05)	2.45 (1.05)	2.19 (1.03)	2.31 (1.05)	2.48 (1.05)	2.39 (1.04)	2.33 (1.08)	1.94 (0.94)	2.66 (1.03)
6. Scientific Community	3.79 (0.88)	3.77 (0.87)	3.83 (0.90)	3.81 (0.86)	3.76 (0.90)	3.76 (0.89)	3.86 (0.82)	3.83 (0.92)	3.79 (0.85)
*General Trust*									
7. Italian Citizens	2.66 (0.71)	2.66 (0.70)	2.64 (0.73)	2.62 (0.68)	2.70 (0.76)	2.67 (0.71)	2.61 (0.68)	2.59 (0.68)	2.70 (0.73)
8. Humankind	2.78 (0.77)	2.76 (0.76)	2.83 (0.80)	2.77 (0.74)	2.80 (0.82)	2.77 (0.78)	2.77 (0.76)	2.76 (0.79)	2.80 (0.86)
*Coping Strategies*									
9. Seeking Social Support	4.25 (1.06)	4.37 (1.06)	3.91 (0.98)	4.28 (1.01)	4.19 (1.28)	4.26 (1.06)	4.21 (1.04)	4.16 (1.03)	4.30 (1.07)
10. Positive Reinterpretation	5.53 (0.87)	5.63 (0.82)	5.27 (0.93)	5.50 (0.91)	5.58 (0.80)	5.51 (0.88)	5.60 (0.83)	5.55 (0.94)	5.52 (0.82)
11. Problem-Solving Orientation	4.45 (0.97)	4.53 (0.97)	4.23 (0.94)	4.38 (0.95)	4.54 (0.98)	4.41 (0.97)	4.55 (0.96)	4.35 (0.95)	4.51 (0.98)
12. Avoidance	2.51 (1.05)	2.44 (1.02)	2.71 (1.11)	2.53 (1.03)	2.49 (1.08)	2.60 (1.10)	2.28 (0.86)	2.53 (1.06)	2.50 (1.04)
13. Transcendent Orientation	2.99 (1.95)	3.13 (1.99)	2.63 (1.81)	2.63 (1.86)	3.57 (1.95)	3.11 (1.95)	2.68 (1.92)	1.67 (1.23)	3.86 (1.85)

**Table 2 ijerph-20-06512-t002:** Correlations among measures of trust and coping strategies.

Measures	2	3	4	5	6	7	8	9	10	11	12	13
1.Trust in Regional Government	0.396 **	0.380 **	0.269 **	0.160 **	0.232 **	0.319 **	0.189 **	0.063	0.150 **	0.051	−0.084 *	−0.075 *
2. Trust in Italian Government	1	0.565 **	0.471 **	0.228 **	0.326 **	0.406 **	0.303 **	0.090 **	0.182 **	0.037	−0.082 *	−0.046
3. Trust in Civil Protection		1	0.369 **	0.238 **	0.422 **	0.268 **	0.305 **	0.106 **	0.228 **	0.123 **	−0.099 **	0.043
4. Trust in European Union			1	0.164 **	0.274 **	0.249 **	0.296 **	0.098 **	0.071 *	0.009	0.017	−0.083 *
5. Trust in Catholic Church				1	0.090 **	0.331 **	0.254 **	0.132 **	0.192 **	0.229 **	−0.117 **	0.521 **
6. Trust in Scientific Community					1	0.107 **	0.322 **	0.122 **	0.183 **	0.105 **	−0.120 **	−0.044
7. Trust in Italian Citizens						1	0.415 **	0.077 *	0.145 **	0.103 **	−0.094 **	0.173 **
8. Trust in Humankind							1	0.117 **	0.184 **	0.169 **	−0.064	0.148 **
9. Seeking Social Support								1	0.217 **	0.424 **	0.022	0.217 **
10. Positive Reinterpretation									1	0.460 **	−0.258 **	0.153 **
11. Problem-Solving Orientation										1	−0.288 **	0.246 **
12. Avoidance											1	0.026
13. Transcendent Orientation												1

Note: * *p* < 0.05; ** *p* < 0.01.

**Table 3 ijerph-20-06512-t003:** Hierarchical regression analysis testing the associations between social and general trust with the coping strategies.

	Seeking Social Support	Positive Reinterpretation	Problem-Solving Orientation	Avoidance	Transcendent Orientation
	***β* [95% CI]**	***β* [95% CI]**	***β* [95% CI]**	***β* [95% CI]**	***β* [95% CI]**
*Model 1*					
Gender	**0.194 *** [0.302, 0.612]**	**0.184 *** [0.222, 0.494]**	**0.130 *** [0.137, 0.423]**	**−0.099 ** [−0.398, −0.071]**	**0.106 *** [0.227, 0.696]**
Age	**−0.084 * [−0.010, −0.001]**	−0.021 [−0.005, 0.003]	0.046 [−0.001, 0.007]	**−0.084 * [−0.011, −0.001]**	**0.087 ** [0.004, 0.019]**
Residence	−0.025 [−0.208, 0.098]	**0.078 * [0.016, 0.259]**	**0.105 ** [0.071, 0.347]**	−0.034 [−0.231, 0.073]	**0.191 *** [0.545, 0.987]**
COVID-19	−0.047 [−0.273, 0.052]	0.033 [−0.061, 0.197]	**0.081 * [0.024, 0.325]**	**−0.133 *** [−0.467, −0.175]**	−0.038 [−0.412, 0.080]
Political Orientation	−0.013 [−0.039, 0.028]	0.033 [−0.051, 0.009]	0.020 [−0.021, 0.38]	**0.078 * [0.004, 0.073]**	**0.101 *** [0.039, 0.142]**
Catholic Religion	0.051 [−0.041, 0.257]	−0.050 [−0.169, 0.090]	0.055 [−0.030, 0.247]	−0.022 [−0.197, 0.099]	**0.495 *** [1.764, 2.196]**
Model 1 *R*^2^	**0.050**	**0.042**	**0.038**	**0.042**	**0.369**
Model 1 *F*	**7.342 *****	**6.170 *****	**5.562 *****	**6.133 *****	**82.12 *****
*Model 2*					
Gender	**0.198 *** [0.311, 0.620]**	**0.170 *** [0.197, 0.464]**	**0.117 *** [0.109, 0.390]**	**−0.098 ** [−0.396, −0.066]**	**0.082 ** [0.144, 0.562]**
Age	**−0.079 * [−0.010, −0.001]**	−0.016 [−0.005, 0.003]	0.035 [−0.002, 0.007]	**−0.083 * [−0.011, −0.001]**	0.041 [−0.001, 0.012]
Residence	0.033 [−0.233, 0.086]	**0.090 * [0.033, 0.284]**	**0.102 ** [0.064, 0.342]**	−0.038 [−0.250, 0.077]	0.157 *** [0.424, 0.831]
COVID-19	−0.049 [−0.273, 0.047]	0.024 [−0.077, 0.171]	**0.082 ** [−0.030, 0.324]**	**−0.127 *** [−0.452, −0.160]**	−0.027 [0.336, 0.108]
Political Orientation	−0.028 [−0.021, 0.049]	−0.006 [−0.033, 0.028]	0.044 [−0.010, 0.049]	0.061 [−0.005, 0.067]	**0.134 *** [0.073, 0.167]**
Catholic Religion	0.015 [−0.125, 0.187]	**−0.084 * [−0.279, −0.013]**	−0.020 [−0.180, 0.104]	0.013 [−0.126, 0.186]	**0.348 *** [1.172, 1.615]**
Trust in Regional Government	−0.023 [−0.134, 0.072]	0.059 [−0.022, 0.145]	0.018 [−0.070, 0.106]	−0.041 [−0.149, 0.049]	**−0.060 * [−0.255, −0.023]**
Trust in Italian Government	0.026 [−0.084, 0.155]	0.049 [−0.034, 0.141]	−0.072 [−0.187, 0.017]	−0.014 [−0.135, 0.100]	**−0.079 * [−0.350, −0.019]**
Trust in Civil Protection	−0.029 [−0.154, 0.081]	0.084 [−0.013, 0.186]	0.057 [−0.028, 0.160]	−0.015 [−0.129, 0.092]	0.024 [−0.092, 0.207]
Trust in European Union	0.028 [−0.064, 0.136]	**−0.093 * [−0.179, −0.014]**	−0.062 [−0.162, 0.019]	**0.110 ** [0.036, 0.247]**	**−0.066 * [−0.280, −0.023]**
Trust in Catholic Church	**0.087 * [0.011, 0.165]**	**0.134 *** [0.051, 0.170]**	**0.188 *** [0.101, 0.243]**	**−0.075 * [−0.239, −0.040]**	**0.386 *** [0.620, 0.820]**
Trust in Scientific Community	**0.095 * [0.016, 0.212]**	**0.099 ** [0.004, 0.195]**	0.073 [−0.007, 0.164]	**−0.115 ** [−0.239, −0.040]**	−0.014 [−0.164, 0.097]
Trust in Italian Citizens	0.014 [−0.107, 0.150]	0.015 [−0.090, 0.128]	0.006 [−0.099, 0.115]	−0.052 [−0.212, 0.057]	0.044 [−0.056, 0.294]
Trust in Humankind	0.063 [−0.030, 0.196]	**0.093 * [0.002, 0.212]**	**0.121 ** [0.052, 0.253]**	−0.003 [−0.115, 0.107]	**0.083 * [0.075, 0.337]**
Model 2 Δ*R*^2^	**0.030**	**0.089**	**0.069**	**0.033**	**0.144**
Model 2 Δ*F*	**3.437 *****	**10.720 *****	**8.046 *****	**3.730 *****	**30.81 *****

Note: Parameters are beta weights. Significant parameters are in bold. Gender was coded 0 = men, and 1 = women. Place of residence was coded 0 = northern regions, and 1 = central–southern regions. Experience with COVID-19 was coded (0 = no; 1 = yes). Catholic religion was coded (0 = no; 1 = yes). * *p* < 0.05, ** *p* < 0.01, *** *p* < 0.001.

## Data Availability

Data available on request from the authors.

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
