# Peer review of "Adaptive Coping Strategies at the Time of COVID-19: The Role of Social and General Trust"

_ijerph, 2023, doi:10.3390/ijerph20156512_

Round 1

Reviewer 1 Report

The current study explored people’s social trust and general trust when they just encountered the COVID-19 pandemic and the relationship between their trust and their coping strategies. This study is meaningful and could evoke us to explore a deeper relationship between trust and coping strategies. But, there are still some questions I want to discuss with the authors.

 1. The article was mainly about trust, coping strategies, and their relationships. Although they are related to people’s well-being, well-being is not the main point of the manuscript. I recommended adjusting the title to point out coping strategies directly.

2. The study collected data from the same period, which was a cross-sectional study. It is not suitable to explore the prediction in a cross-sectional study. Besides, in the current manuscript, the authors used “mediation analysis” in their Results part, but they only used correlation analysis and regression analysis. Although mediation analysis is based on regression analysis, they could explain further or use mediation analysis directly. And, I recommend that they could express the correlation more conservatively, but not use the expression similar to “predict”.

3. In the manuscript, the authors measured and analyzed coping strategies with five critical dimensions. Then why did they introduce other classifications in their Introduction?

4. The authors used a series of pairwise t-tests to explore the role of different factors. Did they adjust their results to eliminate the false from multiple analyses?

5. The most important, results showed that coping strategies had different correlations with different trust. How to explain the different correlations between them? For example, “Problem-Solving Orientation” was significantly correlated with “Trust in Civil Protection” and “Trust in Catholic Church”, but did not have a similar result with “Trust in Regional Government” or “Trust in Italian Government”. Please consider how to explain such results.

Reviewer 2 Report

I would like to say thanks for the opportunity to review this article.

The article presented has an interesting and actual theme with relevance for the improvement of health interventions in future pandemic situations.

Overall, the article has a scientific and appropriate writing, including all the components of a good scientific research. The title and abstract are related with the content. The keywords are linked to the research, and majorly are indexed.

Introduction allows the framing of the theme and the research itself. The main goal is appropriate. Methodology is scientifically appropriate. Inclusion and exclusion criteria of sample are not included

Results are adequate and allow to answer the goals. Some forms of results presentations are not the best to understand and quickly analyse results. The is not clear the statistical test and procedure for correlations and regression analysis. Discussion is done according to the results of the study, allowing a comparison and analysis with the scientific current evidence. Authors present limitations but no strengths and valid conclusions.

References are pertinent, adequate, but being such an important and recent theme, it was important to include more recent references (almost 45% have more than 5 years and more than 35% have more than 10 years).

For this, we suggest the following corrections:

- review the sample criteria presentation

- include procedures and tests used for correlations and regressions

- review the result presentation. Table 1 and 2 have no general explanation, and might be overlapped, or could be joined to 3.2 and 3.3 section of results. Section 3.2 and 3.3 does not have an easy reading and interpretation. The reunion of all data in a table could be an improvement.

Thank you.
